# Organic Matter and Heavy Metal Ions Removal from Surface Water in Processes of Oxidation with Ozone, UV Irradiation, Coagulation and Adsorption

Beata Karwowska * and Elżbieta Sperczyńska

Department of Environmental Engineering and Biotechnology, Faculty of Infrastructure and Environment, Czestochowa University of Technology, Dąbrowskiego St. 69, 42-201 Częstochowa, Poland
* Correspondence: beata.karwowska@pcz.pl; Tel.: +48-34-3250-491; Fax: +48-34-3250-101

**Abstract:** Organic matter present in natural water is a serious problem during water treatment in terms of the possibility of creating disinfection by-products (DBP). The new materials and processes are still analyzed in order to improve the efficient removal of organic matter as well as other pollutants from water intended for human consumption. The aim of the presented study was to evaluate the efficiency of using various combined processes: (a) coagulation and adsorption, (b) oxidation with ozone and coagulation, (c) oxidation with ozone, coagulation and adsorption, and (d) oxidation with ozone, UV irradiation, coagulation and adsorption for treatment of the modified surface water. In the presented study, the changes in pH, turbidity, color, organic matter content (evaluated as oxidizability, total and dissolved organic carbon content, UV 254 and 272 absorbances), and $Ni^{2+}$, $Cd^{2+,}$ and $Pb^{2+}$ ions content were determined during modified surface water purification. Supporting the coagulation process by adsorption with additional ozonation and UV irradiation of the water sample improved the treatment processes of modified surface water. Processes associated with coagulation increased the efficiency of removing color by 4–16% and reduced the value of $UV_{254}$ and $UV_{272}$ absorbance by 10–20%. Using ozonation did not significantly change the content of total organic matter. Heavy metal ions were mostly removed by coagulation and adsorption processes (40–60%). Including ozone, oxidation resulted in insignificant changes in the concentration of metal ions in the purified water (less than 5%). During the treatment of the tested, modified surface water, the best efficiency was observed after the use of coagulation and adsorption processes enhanced with ozone oxidation. The additional involvement of UV irradiation did not have a significant effect on the removal of the analyzed pollutants.

**Keywords:** surface water treatment; coagulation; adsorption; ozonation; UV irradiation; NOM; heavy metals

## 1. Introduction

The largest and most diverse group of water pollutants consists of organic substances (natural organic matter NOM). Their presence is found in all types of waters: the highest concentration–in surface waters, smaller in infiltration waters, and the smallest–in groundwater. Removing NOM during the process of water purification is a priority. In the composition of NOM, it is possible to distinguish two basic groups: non-humic substances consisting of compounds such as amino acids, polysaccharides, hydrocarbons, carbohydrates, fats, waxes, resins, and low molecular organic acids, as well as complex heterogeneous humic substances. Humic substances are formed in the process of microbiological decomposition (humification) of plant and animal residues occurring in soil, sediments, and natural waters. These substances give color to water and participate in transporting micropollutants. Their presence also has a negative impact on the taste and smell of water. Humic substances are believed to constitute half of the total organic carbon present in water [1].

A significant problem in the course of water purification associated with the presence of NOM consists of the possibility of creating disinfection by-products (DBP). The two main groups of DBP are trihalomethanes (THMs) and halogen acetic acids (HAAs), which may occur in waters disinfected with chlorine [2,3]. Therefore, in the case of purifying natural waters containing significant amounts of organic carbon, organic matter should be removed before the disinfection process. The recommended method of removing DBP precursors consists of intensified coagulation and adsorption with activated carbon [4,5]. Coagulation usually results in reducing the content of organic matter with significant molecular weights, including humic acids, whereas the fraction of organic compounds with lower masses are removed in the adsorption processes with activated carbon. Furthermore, activated carbon also adsorbs substances that may constitute DBP precursors. Therefore, the consequence seems to be using a combination of both processes, which should lead to reducing the content of organic matter with a wide range of masses. Usually, powdered activated carbon (PAC) is introduced into the treated water during the coagulation process. The entry point is the rapid mixing chamber. Activated carbon is added with a delay in relation to the coagulant [6].

Coagulation can be supported by the oxidation process, for example, with ozone [7,8]. Ozone is able to react with different substances occurring in natural water in a direct chemical transformation [9]. The other type of chemical process after ozonation can be the indirect reaction of free radicals formed as a result of ozone molecule decomposition in water [10]. Initial ozonation may cause an increase in the average size of colloidal particles, forming colloidal particles from dissolved organic matter and improving the removal of organic matter or turbidity in sedimentation and filtration processes. Initial ozonation can cause changes in the structure of organic compounds present in water, the formation of colloidal particles from dissolved organic matter, and improve turbidity removal in the sedimentation process [9].

Heavy metals (zinc, copper, nickel, lead, cadmium) may also be present in waters. A number of metals tend to form complexes, mainly chelates, with organic matter present in water, which is important in terms of choosing the proper method of their removal. The concentration of heavy metal ions, for example, nickel, cadmium, and lead, are limited in waters intended for consumption by a Directive of the European Union [11] and in Poland by a Regulation of the Minister of Health [12].

Some of the heavy metals, such as zinc or copper in low concentrations, are essential micro-elements in metabolic processes taking place in living organisms, but their increased concentrations and even low content of others (lead, cadmium, mercury) are dangerous for the health of plant organisms, animals, and humans [13,14]. Usually, there are no concentrations of heavy metals in surface waters that exceed the legal limits. However, there have been reports, for example, from Cameroon [15] or China [16], indicating significant water pollution with these elements, which may pose serious threats to the environment. Surface waters in mining areas are at a high risk of elevated concentrations of heavy metals [17]. Metals with water used in production processes can penetrate into surface and groundwater and then can be introduced to the food chain. This problem is particularly visible in developing countries of Africa and Asia [17,18].

Heavy metal ions are removed from the water mainly as a result of coagulation, precipitation, or adsorption. It turns out that the efficiency of removing metals in the coagulation process depends on the type of coagulant used and the general conditions of handling the process. Reducing the content of nickel, cadmium, and lead ions increases along with the pH of the environment. The highest removal of the mentioned metals is achieved in the pH range in which these metals form hardly soluble inorganic connections [19]. The pH range is dependent on the type of precipitated compounds. For example, for hydroxides the ranges are: 10.0 to 10.5 for nickel, 11.0 to 11.5 for cadmium and 9.0 to 9.5 for lead [20].

Adsorption is also a process commonly used to reduce the content of heavy metal ions in water. There is a constant search for new materials, or the activated carbon classically used in the adsorption is modified [21] in order to increase the efficiency of the process [22]

because choosing the correct adsorbent is an essential element of planning a high-efficiency process. Attempts are also made to use waste materials to prepare efficient heavy metal adsorbents [23].

The purpose of the presented studies was to assess the efficiency of removing turbidity and color, natural organic matter, and heavy metal ions from surface water using processes of oxidation (ozonation and UV irradiation), coagulation, and adsorption. The conducted research was intended to evaluate the possibility of supporting the coagulation process with various variants of combination with PAC adsorption, ozone oxidation, and UV irradiation during the modified surface water treatment.

## 2. Research Materials, Course, and Methodology

### 2.1. Materials

The surface water samples, which were collected in December 2021 and February 2022 from the Warta river in Częstochowa (Poland), were used for the research. The tests were carried out after modifying the concentration of heavy metal ions in the surface water. The modification consisted in introducing heavy metal ions in the form of nitrate(V) solutions into natural water, in an amount ensuring the total concentration of ions of each metal approx. 0.4 mg/L. For this purpose, working solutions of nickel(II), cadmium(II), and lead(II) nitrates(V) at a concentration of 1 mg/mL were prepared. Then, given amounts of metal solutions were introduced depending on the volume of the water sample.

As coagulants, pre-hydrolyzed polyaluminum chlorides with the trade name PAX produced by KEMIPOL company in Police (Poland), the characteristics of which are presented in Table 1, were used. Pre-hydrolyzed coagulants contain hydroxyl groups that determine their increased alkalinity. PAX18 is a low-alkalinity coagulant, PAX-XL10-medium-alkalinity, while PAX-XL19H and PAX-XL19F are high-alkalinity coagulants. For the tests, working solutions of coagulants containing 1 mg Al in 1 mL of solution were prepared by properly diluting the commercial products.

**Table 1.** Characteristics of the polyaluminum coagulants.

| Parameter | Unit | Coagulant | | | |
|---|---|---|---|---|---|
| | | PAX18 | PAX-XL10 | PAX-XL19H | PAX-XL19F |
| Basicity | % | $41 \pm 3$ | $65 \pm 5$ | $85 \pm 5$ | $85 \pm 5$ |
| $OH^-/Al^{3+}$ | - | 1.23 | 1.95 | 2.55 | 2.55 |
| Aluminum | % | $9.0 \pm 0.3$ | $5 \pm 0.2$ | $12.5 \pm 0.5$ | $8.5 \pm 0.3$ |
| Chlorides | % | $21.0 \pm 1.0$ | $11.5 \pm 1.0$ | $8.5 \pm 1$ | $5.5 \pm 0.5$ |
| pH | - | <1.0 | $3.0 \pm 0.5$ | $3.5 \pm 0.4$ | $4.0 \pm 0.5$ |

As adsorbents, powdered activated carbons with the trade names CWZ-22 and AKPA-22 produced by GRYFSKAND company in Hajnówka (Poland) were used, the characteristics of which are presented in Table 2.

**Table 2.** Selected properties of powdered activated carbons (PAC).

| Properties | Unit | Powdered Activated Carbon | |
|---|---|---|---|
| | | CWZ-22 | AKPA-22 |
| Specific surface area | $m^2/g$ | 960 | 914 |
| Iodine number | mg/g | 1032 | 949 |
| Methylene number | $cm^3$ | 29 | 28 |
| Granulation < 0.06 mm | % | 93 | 98 |

### 2.2. Course of Research and Analysis Methodology

The research was carried out in four stages under laboratory conditions. During the first stage, the oxidation process was carried out using water ozonation and UV irradiation. The ozone was produced by a laboratory generator (L20 SPALAB model by Korona) from

high-purity oxygen (99%). A working volume of 1.7 L surface water was measured to three laboratory reactors with a volume of 2 L, and the water was ozonated successively over 5, 10, and 15 min. A spherical stone diffuser at the bottom of the reactors was used to disperse the ozone. All experiments were carried out at a constant gas flow rate of 1.5 L/min, and the dose of ozone introduced into the water over 1 min was about 1.2 mg $O_3$/L. Unreacted ozone in the waste gas was discharged into a potassium iodide solution. Water ozonation in combination with UV irradiation was also carried out for a duration of 10 min using the HERAEUS Group low-pressure immersion lamp generating light with a wavelength of 254 nm. After ending the oxidation process, as well as oxidation and irradiation, the water samples were analyzed.

During the second stage, the coagulation process was carried out. To do this, four coagulants, each in doses of 1.5; 2.5; 3.5, and 4.5 mg Al/L, were introduced into 2 L beakers with 1.7 L of water. With the use of mechanical mixers, a quick stirring was carried out for one minute (at 300 rpm) and then a slow stirring for 10 min (30 rpm). After this time, the samples were left to settle for 60 min. Then 0.3 L of water was decanted, and the samples were analyzed.

The adsorption process was carried out during the third stage. For this purpose, 0.2 L of water and two types of PAC were measured for 8 conical flasks, each in doses of 30, 60, 90, and 120 mg/L. The water with PAC was shaken at a frequency of 550 rpm for 30 min. After shaking, the liquid samples were filtered through filter paper (Munktel filter paper grade 389). The resulting filtrates were analyzed.

During the fourth stage, 1.7 L of water was measured in each beaker with a volume of 2 L: non-ozonated water to two beakers, water ozonated for 10 min to two following beakers, and water ozonated and irradiated with UV irradiation to the last beaker (Figure 1). The coagulant selected in the second stage was introduced into the water in all beakers at a dose of 4 mg Al/L. The beakers were placed on a station with stirrers, and quick stirring was carried out over 1 min using 300 rpm. Then, PAC selected in stage three was added to the water in beakers no. 2, 4, and 5 at a dose of 100 mg/L, and the mixing of water in all beakers was continued for 2 min (Figure 1). After the rapid mixing, the rotational speed was reduced to 30 rpm, and the mixing was continued for 10 min. The samples were left in the beakers for a 60-min settling time. Then 0.3 L of water was decanted from each beaker. For both modified and purified surface water in each of the four stages, the following determinations were made: pH, turbidity, color, oxidizability (OXI), total and dissolved organic carbon (TOC and DOC), $UV_{254}$ and $UV_{272}$ absorbance, as well as the concentration of heavy metal ions.

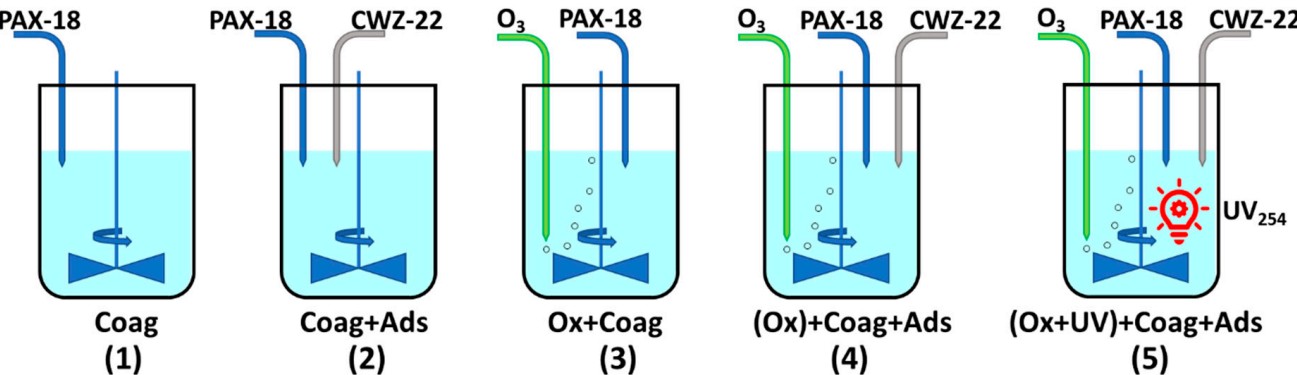

**Figure 1.** Scheme of the fourth stage of studies, Coag–coagulation; Coag + Ads–coagulation and adsorption; Ox + Coag–ozonation and coagulation; Ox + Coag + Ads–ozonation and coagulation + adsorption; (Ox + UV) + Coag + Ads–ozone oxidation + UV irradiation and coagulation and adsorption.

The analyses were carried out under ambient conditions, at a temperature of 20 °C, using the following methods: pH–potentiometric (pHmeter HI2002 by Hanna Instruments), turbidity–nephelometric (HI 98703 turbidity meter by Hanna Instruments), color–

comparing with standards of the platinum-cobalt scale, oxidizability–$KMnO_4$ titration, TOC and DOC (after water filtration through a 0.45 μm membrane filter)–through high-temperature oxidation in an oxygen stream and $CO_2$ measurement using an NDIR detector (Vario TOC cube organic carbon analyzer by Elementar), absorbance in UV ultraviolet at a wavelength of 254 nm and 272 nm (1 cm cuvette) using a UV 5600 Shanghai spectrophotometer, China Metash Instruments, the concentration of heavy metal ions (nickel: $Ni^{2+}$, lead: $Pb^{2+}$ and cadmium: $Cd^{2+}$) using the atomic absorption spectrometry method (novAA 400 spectrometer by Analytik Jena) with flame atomization. The metal cations concentration was detected from the standard calibration curve prepared with the use of standard metal solutions from Sigma–Aldrich. The analyses were performed in three repetitions, according to standard methods and recommendations for water intended for human consumption [11,12,24].

## 3. Results and Discussion

Surface water samples collected in December 2021and February 2022 were characterized by turbidity of 6.9–12.8 NTU and color of 55–65 mg Pt/L. The water pH was slightly alkaline (pH value varied in the range of 7.08–7.95). The content of organic compounds determined by the OXI and TOC indicators was 7.3–8.9 mg $O_2$/L and 7.1–8.9 mg/L. The $UV_{254}$ and $UV_{272}$ absorbance was equal to 0.298–0.365 and 0.245–0.296 1/cm, respectively.

The concentration of the analyzed heavy metal ions in the surface water before modification was very low. For nickel and lead, it was 0.070 and 0.001 mg/L, respectively, while for cadmium, it was below the limit of quantification. Therefore, analyzing changes in metal concentrations as a result of single and combined water treatment processes required increasing the concentration of these metals under laboratory conditions. The determined concentrations of nickel, cadmium and lead in the modified surface water ranged from 0.38 to 0.40 mg/L.

### 3.1. Water Ozonation

The impact of the time of ozonation on changing the turbidity and color of water, concentrations of TOC, DOC, heavy metal ions, and the values of oxidizability, as well as $UV_{254}$ and $UV_{272}$ absorbance, are presented in Table 3. The effect of ozonation of water with UV irradiation on the mentioned water quality parameters is also shown in Table 3.

**Table 3.** Characteristic parameters of the modified surface water and water after oxidation with ozone (ozonation) and UV irradiation processes.

| Parameter | Unit | Modified Surface Water | Treated Water | | | |
|---|---|---|---|---|---|---|
| | | | Ozonation–$O_3$ | | | $O_3$ + UV Irradiation |
| | | | 5 min | 10 min | 15 min | 10 min |
| pH | - | 7.95 ± 0.01 | 8.12 ± 0.02 | 8.07 ± 0.01 | 8.03 ± 0.01 | 8.01 ± 0.02 |
| Turbidity | NTU | 6.89 ± 0.03 | 5.91 ± 0.02 | 5.36 ± 0.02 | 5.13 ± 0.02 | 5.11 ± 0.02 |
| Colour | mg Pt/L | 60 ± 3 | 32 ± 3 | 25 ± 2 | 20 ± 2 | 22 ± 2 |
| Absorbance 254 nm 272 nm | 1/cm | 0.298 ± 0.011 0.245 ± 0.012 | 0.188 ± 0.009 0.140 ± 0.010 | 0.116 ± 0.004 0.078 ± 0.010 | 0.071 ± 0.009 0.055 ± 0.007 | 0.049 ± 0.006 0.035 ± 0.008 |
| OXI * | mg $O_2$/L | 7.3 ± 0.2 | 6.2 ± 0.1 | 5.8 ± 0.1 | 5.3 ± 0.1 | 5.4 ± 0.1 |
| TOC * | mg C/L | 7.24 ± 0.05 | 7.08 ± 0.05 | 6.71 ± 0.03 | 6.52 ± 0.04 | 5.96 ± 0.02 |
| DOC * | mg C/L | 7.04 ± 0.04 | 6.96 ± 0.04 | 6.58 ± 0.04 | 6.46 ± 0.02 | 5.70 ± 0.02 |
| Nickel | mg/L | 0.38 ± 0.01 | 0.34 ± 0.02 | 0.34 ± 0.02 | 0.34 ± 0.01 | 0.34 ± 0.01 |
| Cadmium | mg/L | 0.39 ± 0.02 | 0.36 ± 0.01 | 0.35 ± 0.01 | 0.35 ± 0.02 | 0.34 ± 0.01 |
| Lead | mg/L | 0.38 ± 0.03 | 0.36 ± 0.03 | 0.34 ± 0.02 | 0.34 ± 0.02 | 0.33 ± 0.01 |

Notes: * OXI (oxidizability), TOC (total organic carbon), DOC (dissolved organic carbon).

A slight reduction in water turbidity from 14 to 26% has been determined in terms of using ozonation, depending on its duration (5–15 min), whereas an advantage resulting from the decomposition of humic substances was reducing the intensity of the color of water subjected to ozonation at 47–67%. Supporting ozonation with UV irradiation had a small impact on improving the effectiveness of removing turbidity and color.

The value of the oxidizability index in water after ozonation decreased by 15–27%, and the TOC content by 2–10%. This means that the ozonation process slightly changed the total content of organic matter in water, which is consistent with the research conducted by Chiang et al. [9]. The combined effect of ozone with UV irradiation increased the reduction of TOC content by an additional 11%. According to Zainudin et al. [25], ozone oxidation generally increases the biodegradability of NOM in water by converting larger molecules of organic compounds into smaller ones that are more readily biodegradable. The best efficiency was obtained in the case of $UV_{254}$ and $UV_{272}$ absorbance, the values of which decreased in ozonated water by 37–78% and after using ozonation with UV irradiation by 84–86%. A significant decrease in the value of UV absorbance indicates that the qualitative composition of organic compounds, a fraction of organic carbon characterized by a high content of aromatic compounds, is changing [26]. $UV_{254}$ absorbance is an indicator of the content of aromatic rings, considered highly reactive compounds during water disinfection, and thus used to assess the content of DBP formation precursors. The measurement of $UV_{272}$ absorbance is also used for this purpose [27]. Therefore, as a result of using ozonation or ozonation with UV irradiation as an initial stage of water treatment at the beginning of the technological system, it is possible to significantly reduce the content of this part of organic matter, which is responsible for the formation of DBP (for example THM) during water disinfection with chlorine [25].

As a result of oxidation with ozone as well as oxidation with UV irradiation, a slight decrease in the content of the analyzed heavy metals was observed (Table 3). In the case of nickel, the concentration of this metal decreased by about 10% and basically did not depend on the ozone dose introduced during the oxidation process. Also, adding UV irradiation to oxidation did not affect the changes in the amount of $Ni^{2+}$ ions removed. For cadmium and lead, the measured concentrations decreased as the ozonation time increased and the process extended with UV irradiation. However, the changes were not significant and oscillated within the accuracy limits of the method of determination. Therefore, the ozonation time (10 min) selected on the basis of the course of changes in parameters related to the content of organic matter in the tested water was used for further studies.

### 3.2. Coagulation

The results of the impact of the type and dose of the coagulant on changing the turbidity and color of water, TOC, DOC, heavy metal ions, and the values of oxidability, $UV_{254}$ and $UV_{272}$ absorbance are presented in Tables 4 and 5 and Figure 2.

**Table 4.** Characteristic parameters of the modified surface water and water after the coagulation process with PAX18 and PAX-XL19H.

| Parameter | Unit | Modified Surface Water | Treated Water | | | | | | | |
|---|---|---|---|---|---|---|---|---|---|---|
| | | | PAX18 Dose, mg/L | | | | PAX-XL19H Dose, mg/L | | | |
| | | | 1.5 | 2.5 | 3.5 | 4.5 | 1.5 | 2.5 | 3.5 | 4.5 |
| pH | - | 7.64 ± 0.02 | 7.28 ± 0.01 | 7.14 ± 0.01 | 7.02 ± 0.01 | 6.94 ± 0.02 | 7.54 ± 0.01 | 7.65 ± 0.03 | 7.68 ± 0.01 | 7.66 ± 0.01 |
| Turbidity | NTU | 8.46 ± 0.03 | 6.05 ± 0.02 | 3.04 ± 0.02 | 1.39 ± 0.01 | 1.28 ± 0.01 | 4.49 ± 0.02 | 2.92 ± 0.02 | 2.16 ± 0.02 | 1.47 ± 0.01 |
| Colour | mg Pt/L | 65 ± 3 | 40 ± 3 | 20 ± 2 | 13 ± 1 | 10 ± 1 | 35 ± 2 | 27 ± 2 | 18 ± 1 | 14 ± 1 |
| Absorbance 254 nm 272 nm | 1/cm | 0.314 ± 0.020 0.260 ± 0.020 | 0.173 ± 0.002 0.139 ± 0.020 | 0.140 ± 0.009 0.113 ± 0.009 | 0.106 ± 0.010 0.089 ± 0.010 | 0.090 ± 0.009 0.073 ± 0.011 | 0.201 ± 0.020 0.171 ± 0.008 | 0.150 ± 0.012 0.124 ± 0.008 | 0.133 ± 0.010 0.106 ± 0.008 | 0.114 ± 0.009 0.091 ± 0.008 |
| OXI | mg $O_2$/L | 7.9 ± 0.2 | 6.5 ± 0.1 | 4.8 ± 0.1 | 3.9 ± 0.1 | 3.1 ± 0.1 | 6.1 ± 0.2 | 5.6 ± 0.1 | 4.3 ± 0.1 | 3.8 ± 0.1 |
| TOC | mg C/L | 7.38 ± 0.05 | 6.04 ± 0.04 | 4.66 ± 0.02 | 3.77 ± 0.03 | 3.04 ± 0.02 | 5.69 ± 0.04 | 4.78 ± 0.05 | 4.19 ± 0.03 | 3.71 ± 0.04 |
| DOC | mg /L | 6.96 ± 0.04 | 5.76 ± 0.02 | 4.37 ± 0.04 | 3.53 ± 0.05 | 2.87 ± 0.03 | 5.57 ± 0.03 | 4.74 ± 0.02 | 4.14 ± 0.04 | 3.63 ± 0.03 |
| Aluminum | mg Al/L | 0.01 ± 0.01 | 0.14 ± 0.02 | 0.08 ± 0.01 | 0.05 ± 0.01 | 0.05 ± 0.01 | 0.02 ± 0.01 | 0.02 ± 0.01 | <0.01 | <0.01 |
| Nickel | mg/L | 0.38 ± 0.01 | 0.29 ± 0.01 | 0.26 ± 0.01 | 0.24 ± 0.01 | 0.23 ± 0.01 | 0.31 ± 0.02 | 0.29 ± 0.02 | 0.26 ± 0.01 | 0.25 ± 0.01 |
| Cadmium | mg/L | 0.39 ± 0.02 | 0.29 ± 0.01 | 0.26 ± 0.01 | 0.22 ± 0.01 | 0.21 ± 0.01 | 0.30 ± 0.02 | 0.27 ± 0.02 | 0.24 ± 0.01 | 0.24 ± 0.01 |
| Lead | mg/L | 0.38 ± 0.03 | 0.21 ± 0.01 | 0.19 ± 0.01 | 0.17 ± 0.01 | 0.17 ± 0.02 | 0.26 ± 0.01 | 0.25 ± 0.01 | 0.23 ± 0.01 | 0.23 ± 0.02 |

**Table 5.** Characteristic parameters of the modified surface water and water after the coagulation process with PAX-XL10 and PAX-XL19F.

| Parameter | Unit | Modified Surface Water | Treated Water | | | | | | | |
| --- | --- | --- | --- | --- | --- | --- | --- | --- | --- | --- |
| | | | PAX-XL10 Dose, mg/L | | | | PAX-XL19F Dose, mg/L | | | |
| | | | 1.5 | 2.5 | 3.5 | 4.5 | 1.5 | 2.5 | 3.5 | 4.5 |
| pH | - | 7.26 ± 0.01 | 7.25 ± 0.01 | 7.22 ± 0.02 | 7.19 ± 0.01 | 7.13 ± 0.01 | 7.39 ± 0.02 | 7.43 ± 0.01 | 7.42 ± 0.01 | 7.41 ± 0.02 |
| Turbidity | NTU | 12.30 ± 0.08 | 8.22 ± 0.04 | 5.78 ± 0.04 | 3.62 ± 0.02 | 2.52 ± 0.04 | 6.45 ± 0.04 | 4.82 ± 0.02 | 1.93 ± 0.02 | 1.64 ± 0.01 |
| Colour | mg Pt/L | 55 ± 3 | 40 ± 3 | 35 ± 3 | 25 ± 2 | 18 ± 1 | 40 ± 3 | 30 ± 2 | 20 ± 2 | 15 ± 1 |
| Absorbance 254 nm 272 nm | 1/cm | 0.365 ± 0.005 0.296 ± 0.008 | 0.253 ± 0.009 0.198 ± 0.007 | 0.191 ± 0.010 0.155 ± 0.011 | 0.166 ± 0.007 0.131 ± 0.010 | 0.144 ± 0.006 0.113 ± 0.008 | 0.225 ± 0.012 0.182 ± 0.006 | 0.207 ± 0.008 0.165 ± 0.009 | 0.161 ± 0.007 0.127 ± 0.006 | 0.148 ± 0.006 0.118 ± 0.008 |
| OXI | mg $O_2$/L | 8.9 ± 0.2 | 7.8 ± 0.1 | 7.3 ± 0.1 | 6.0 ± 0.2 | 4.9 ± 0.1 | 7.0 ± 0.1 | 6.5 ± 0.2 | 5.3 ± 0.1 | 4.5 ± 0.1 |
| TOC | mg C/L | 8.85 ± 0.05 | 6.65 ± 0.04 | 6.04 ± 0.04 | 5.10 ± 0.07 | 4.63 ± 0.06 | 6.23 ± 0.03 | 5.62 ± 0.08 | 4.87 ± 0.08 | 4.34 ± 0.04 |
| DOC | mg C/L | 8.05 ± 0.04 | 6.41 ± 0.06 | 5.78 ± 0.08 | 4.88 ± 0.08 | 4.37 ± 0.05 | 6.08 ± 0.06 | 5.35 ± 0.06 | 4.57 ± 0.07 | 4.20 ± 0.02 |
| Aluminum | mg Al/L | 0.01 ± 0.01 | 0.16 ± 0.2 | 0.15 ± 0.02 | 0.12 ± 0.2 | 0.10 ± 0.01 | 0.08 ± 0.01 | 0.06 ± 0.01 | <0.01 | <0.01 |
| Nickel | mg/L | 0.37 ± 0.01 | 0.35 ± 0.01 | 0.34 ± 0.01 | 0.32 ± 0.01 | 0.28 ± 0.01 | 0.28 ± 0.02 | 0.27 ± 0.01 | 0.25 ± 0.01 | 0.25 ± 0.01 |
| Cadmium | mg/L | 0.40 ± 0.01 | 0.33 ± 0.01 | 0.30 ± 0.01 | 0.29 ± 0.02 | 0.27 ± 0.01 | 0.30 ± 0.01 | 0.30 ± 0.02 | 0.27 ± 0.01 | 0.27 ± 0.01 |
| Lead | mg/L | 0.38 ± 0.03 | 0.33 ± 0.01 | 0.30 ± 0.02 | 0.27 ± 0.01 | 0.25 ± 0.02 | 0.27 ± 0.01 | 0.27 ± 0.01 | 0.27 ± 0.01 | 0.24 ± 0.03 |

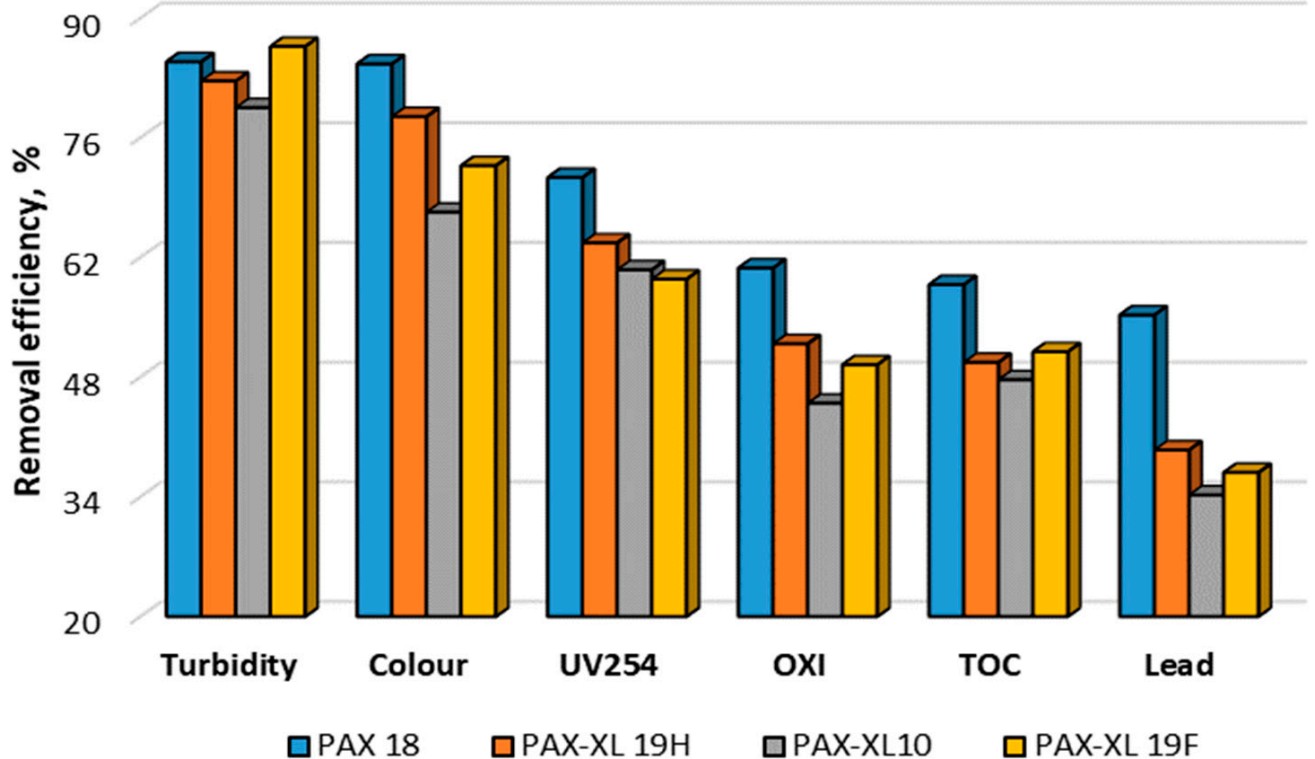

**Figure 2.** Changes in the value of selected parameters of the modified surface water treated in the coagulation process with different coagulants aid (dose 4.5 mg Al/L); UV254–absorbance at wavelength 254 nm, OXI–oxidizability, TOC–total organic carbon.

As the dose of coagulants increased, the effect of water purification also increased. The best results were obtained for a dose of 4.5 mg Al/L. These were significant effects in relation to doses of 1.5 and 2.5 mg Al/L while differing little from those achieved at a dose of coagulants of 3.5 mg Al/L. Depending on the used dose, the turbidity was reduced from 28 to 87% and the color of the water from 27 to 85%. In the case of the highest dose of coagulants, the best removal of turbidity was obtained both when using low- and high-alkaline coagulants. In the case of color, the best effects were obtained with the use of low-alkaline. The lowest concentration of aluminum remaining after coagulation was found in the water during the application of PAX19F.

The best result of reducing organic compounds in water was obtained using low-alkaline PAX18 (Figure 2). Reducing the OXI and TOC values at the highest coagulant dose was respectively 61 and 59%, and the $UV_{254}$ and $UV_{272}$ absorbance was 71 and 72%. According to Matilainen et al. [1], aromatic NOM fractions can be more easily removed in the coagulation process compared to other NOM fractions. When using other coagulants,

the efficiency of removing organic matter was 45–61%, 48–59%, and 59–65% for OXI, TOC, and UV absorbance indicators, respectively.

The use of polyaluminum chlorides to remove organic substances was researched by Zhang et al. [28]. They used polyaluminum chlorides with an alkalinity of 63 to 90%. They obtained the best results of lowering the value of $UV_{254}$ absorbance with the use of a coagulant with the lowest alkalinity and the content of DOC depending on the dose both when using a coagulant with the lowest and the highest alkalinity. Depending on the alkalinity, the coagulant includes a different content of monomeric, polymeric, and colloidal forms of aluminum. Aluminum polymer forms are considered stable, while monomer forms can undergo transformation. Under certain conditions, this may result in the formation of in-situ aluminum polymers neutralizing the load of impurities and thus allowing for obtaining proper water purification efficiency with the use of coagulants with low alkalinity containing significant amounts of monomeric forms.

The analysis of obtained results showed that for all analyzed metals, a decrease in the concentration of metal ions was observed with an increase in the dose of the used coagulant (Tables 4 and 5). The PAX18 and PAX19H coagulants showed the highest efficiency. The degree of removing metals using PAX18 varied from 24 to 42% for nickel, 26–46% for cadmium, and 45–55% for lead. In the case of PAX19H, the level of removing individual metals was slightly lower and amounted to 18–34%, 23–38%, and 32–39%, respectively, for nickel, cadmium, and lead. In the case of using other coagulants, the efficiency of removing heavy metals from the water was significantly lower. Therefore, PAX18 was selected as the most optimal coagulant for removing metals from water. In the case of this coagulant, a decrease in concentration was observed for all the analyzed metals, with a dose increase from 1.5 to 3.5 mg Al/L. Increasing the dose to 4.5 mg, Al/L resulted only in slightly increasing the amount of metal removed from the water. Also, taking into consideration the effectiveness of reducing the value of other parameters during coagulation, it was decided to finally use PAX18 at a dose of 4.0 mg Al/L in further research.

### 3.3. Adsorption

The results of the impact of the type and dose of PAC on changing the water color, the concentration of TOC, DOC, and heavy metal ions, as well as the values of oxidizability, $UV_{254,}$ and $UV_{272}$ absorbance, are presented in Table 6.

**Table 6.** Characteristic parameters of the modified surface water and water after adsorption with powdered activated carbon AKPA-22 and CWZ-22.

| Parameter | Unit | Modified Surface Water | Treated Water | | | | | | | |
|---|---|---|---|---|---|---|---|---|---|---|
| | | | AKPA-22 Dose, mg/L | | | | CWZ-22 Dose, mg/L | | | |
| | | | 30 | 60 | 90 | 120 | 30 | 60 | 90 | 120 |
| pH | - | 7.74 ± 0.02 | 7.28 ± 0.01 | 7.14 ± 0.02 | 7.02 ± 0.02 | 6.94 ± 0.01 | 7.54 ± 0.02 | 7.65 ± 0.01 | 7.68 ± 0.01 | 7.66 ± 0.01 |
| Colour | mg Pt/L | 65 ± 3 | 40 ± 3 | 40 ± 3 | 37 ± 3 | 30 ± 2 | 37 ± 3 | 35 ± 3 | 27 ± 2 | 22 ± 2 |
| Absorbance 254 nm | 1/cm | 0.318 ± 0.011 | 0.316 ± 0.011 | 0.296 ± 0.008 | 0.293 ± 0.006 | 0.286 ± 0.011 | 0.277 ± 0.008 | 0.270 ± 0.010 | 0.232 ± 0.009 | 0.212 ± 0.004 |
| 272 nm | | 0.254 ± 0.011 | 0.252 ± 0.011 | 0.240 ± 0.006 | 0.236 ± 0.008 | 0.227 ± 0.010 | 0.223 ± 0.006 | 0.219 ± 0.011 | 0.186 ± 0.006 | 0.171 ± 0.006 |
| OXI | mg $O_2$/L | 7.9 ± 0.05 | 7.7 ± 0.2 | 7.6 ± 0.1 | 7.4 ± 0.2 | 6.9 ± 0.1 | 7.4 ± 0.2 | 7.3 ± 0.1 | 6.5 ± 0.1 | 5.9 ± 0.1 |
| TOC | mg C/L | 7.91 ± 0.06 | 7.12 ± 0.10 | 6.88 ± 0.04 | 6.64 ± 0.06 | 6.44 ± 0.06 | 6.68 ± 0.04 | 6.42 ± 0.08 | 6.17 ± 0.04 | 5.87 ± 0.06 |
| DOC | mg C/L | 7.22 ± 0.09 | 6.48 ± 0.10 | 6.21 ± 0.08 | 6.11 ± 0.07 | 5.94 ± 0.08 | 6.14 ± 0.02 | 5.87 ± 0.07 | 5.63 ± 0.02 | 5.42 ± 0.09 |
| Nickel | mg/L | 0.38 ± 0.01 | 0.22 ± 0.01 | 0.19 ± 0.01 | 0.18 ± 0.02 | 0.18 ± 0.01 | 0.24 ± 0.01 | 0.23 ± 0.01 | 0.21 ± 0.02 | 0.21 ± 0.01 |
| Cadmium | mg/L | 0.39 ± 0.02 | 0.21 ± 0.02 | 0.19 ± 0.01 | 0.18 ± 0.01 | 0.16 ± 0.02 | 0.22 ± 0.02 | 0.21 ± 0.01 | 0.18 ± 0.01 | 0.17 ± 0.01 |
| Lead | mg/L | 0.38 ± 0.01 | 0.21 ± 0.01 | 0.18 ± 0.01 | 0.17 ± 0.02 | 0.16 ± 0.01 | 0.20 ± 0.01 | 0.20 ± 0.03 | 0.18 ± 0.01 | 0.17 ± 0.01 |

The adsorption process was carried out using two types of PAC named AKPA-22 and CWZ-22. It was found that as the dose of PAC increased, the effect of water purification also increased. For the highest dose equal to 120 mg/L, a 13 and 19% reduction in the content of organic substances designated as OXI and TOC were obtained in the case of using AKPA-22 carbon, as well as by 25 and 26% during adsorption with CWZ-22 carbon. Removing color for the highest dose of AKPA-22 and CWZ-22 was 54 and 66%, respectively. The UV absorbance also decreased with the use of a higher dose of PAC. The absorbance value was reduced by 33% for both $UV_{254}$ and $UV_{272}$ using the highest dose of CWZ-22 carbon. Lower efficiency was recorded using the AKPA-22 carbon–10 and 11%.

In the case of both adsorbents, it was observed that changing the dose of the used material in the range of 30 to 120 mg/L resulted in decreasing the concentration of metal in the purified water. When changing the dose from 90 to 120 mg/L, the decrease in metal concentration was negligible. For the analyzed water samples, the CWZ-22 carbon showed better effects in terms of removing selected heavy metals. In this case, the efficiency of removing metals from water ranged from 42 to 53% for nickel, 46–59% for cadmium, and 45–58% for lead. Therefore, in the combined processes, it was decided to use the CWZ-22 activated carbon at a dose of 100 mg/L of purified water.

### 3.4. Ozonation, Coagulation, Adsorption

The results of supporting coagulation with the ozonation process, UV irradiation, and adsorption with powdered activated carbon to reduce turbidity, water color, the concentration of TOC, DOC, heavy metal ions, as well as the values of oxidizability, and $UV_{254}$ and $UV_{272}$ absorbance are presented in Table 7 and Figure 3. Coagulation carried out with the use of PAX18 at a dose of 4 mg Al/L allowed removing turbidity by 89%, colour by 78%, and organic compounds by 66% marked as $UV_{254}$ absorbance, 56% as oxidation, 44% as TOC, and 40% as DOC.

**Table 7.** Characteristic parameters of the modified surface water and treated water in the coagulation process and hybrid processes: ozonation (10 min), coagulation (PAX-18 dose 4 mg Al/L), adsorption (CWZ-22 dose 100 mg/L), UV irradiation (10 min).

| Parameter | Unit | Modified Surface Water | Treated Water | | | | |
|---|---|---|---|---|---|---|---|
| | | | Coag | Coag + Ads | Ox + Coag | Ox + Coag + Ads | (Ox + UV) + Coag + Ads |
| pH | - | 7.08 ± 0.01 | 6.75 ± 0.01 | 6.80 ± 0.03 | 6.84 ± 0.01 | 6.84 ± 0.01 | 6.82 ± 0.01 |
| Turbidity | NTU | 12.80 ± 0.03 | 1.39 ± 0.02 | 2.08 ± 0.02 | 1.28 ± 0.02 | 2.62 ± 0.03 | 1.33 ± 0.02 |
| Colour | mg Pt/L | 55 ± 3 | 12 ± 1 | 5 ± 1 | 10 ± 1 | 3 ± 1 | 5 ± 1 |
| Absorbance 254 nm | 1/cm | 0.354 ± 0.020 | 0.120 ± 0.010 | 0.084 ± 0.010 | 0.103 ± 0.010 | 0.052 ± 0.010 | 0.050 ± 0.010 |
| 272 nm | | 0.289 ± 0.020 | 0.099 ± 0.009 | 0.070 ± 0.009 | 0.083 ± 0.010 | 0.041 ± 0.009 | 0.039 ± 0.008 |
| OXI | mg $O_2$/L | 8.9 ± 0.2 | 3.9 ± 0.1 | 3.6 ± 0.1 | 3.4 ± 0.1 | 3.3 ± 0.1 | 3.4 ± 0.1 |
| TOC | mg C/L | 8.85 ± 0.05 | 4.97 ± 0.04 | 4.64 ± 0.05 | 4.56 ± 0.05 | 4.42 ± 0.06 | 4.47 ± 0.04 |
| DOC | mg C/L | 8.05 ± 0.05 | 4.79 ± 0.06 | 4.47 ± 0.05 | 4.39 ± 0.04 | 4.21 ± 0.05 | 4.43 ± 0.04 |
| Aluminum | mg Al/L | <0.01 | 0.05 ± 0.01 | 0.02 ± 0.01 | 0.02 ± 0.01 | <0.01 | <0.01 |
| Nickel | mg/L | 0.37 ± 0.02 | 0.22 ± 0.02 | 0.18 ± 0.01 | 0.21 ± 0.01 | 0.18 ± 0.01 | 0.20 ± 0.01 |
| Cadmium | mg/L | 0.40 ± 0.01 | 0.23 ± 0.01 | 0.18 ± 0.02 | 0.22 ± 0.02 | 0.17 ± 0.01 | 0.23 ± 0.01 |
| Lead | mg/L | 0.38 ± 0.02 | 0.19 ± 0.01 | 0.14 ± 0.02 | 0.18 ± 0.01 | 0.14 ± 0.02 | 0.18 ± 0.02 |

Notes: Coagulation (Coag), coagulation and adsorption (Coag + Ads), oxidation with ozone and coagulation (Ox + Coag), oxidation with ozone and coagulation and adsorption (Ox + Coag + Ads), oxidation with ozone+ UV irradiation and coagulation and adsorption ((Ox + UV) + Coag + Ads).

The turbidity removal efficiency ranged from 80 to 90%. The greatest turbidity decrease (90%) was observed after the application of the initial ozonation before the coagulation process. The involving of the adsorption process using PAC for coagulation, without and with ozonation, resulted in a slight deterioration of turbidity removal compared to coagulation alone due to the penetration of fine PAC particles into the treated water. Samples were decanted and not passed through the filter paper. However, it should be noted that in practice, under technical conditions, the coagulation process is always followed by a filtration process in which PAC particles are separated from the purified water.

Supporting coagulation with adsorption using the CWZ-22 carbon resulted in an additional 12% reduction in color while supporting coagulation by ozonation by 4%. The efficiency of removing color increased by 16% when supporting the coagulation process with ozonation and adsorption processes at the same time. Additionally, introducing the process of UV irradiation did not cause a noticeable reduction in the content of organic substances designated as OXI, TOC, or DOC.

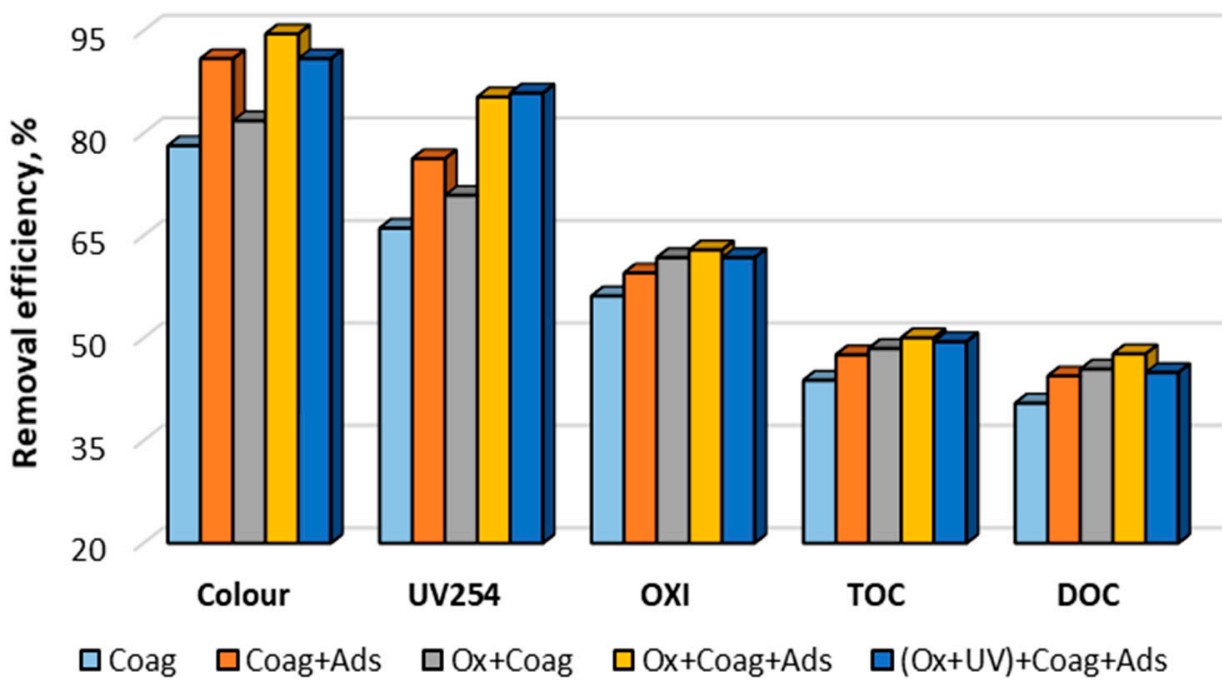

**Figure 3.** Selected parameters removal efficiency during coagulation (Coag), coagulation and adsorption (Coag + Ads), ozone oxidation and coagulation (Ox + Coag), ozone oxidation and coagulation and adsorption (Ox + Coag + Ads), ozone oxidation + UV irradiation and coagulation + adsorption ((Ox + UV) + Coag + Ads); UV254–absorbance at wavelength 254 nm, OXI–oxidizability, TOC–total organic carbon DOC–dissolved organic carbon.

In the case of organic compounds, supporting the coagulation with adsorption or ozone oxidation resulted in further reducing TOC and DOC by 4 and 5%, respectively. The reduction in the value of $UV_{254}$ and $UV_{272}$ absorbance in the process of coagulation combined with adsorption was 76%, and when it was preceded by ozonation, then it amounted to 71%. Carrèire et al. [4] used combining the coagulation process with adsorption with PAC for the surface water treatment (the mixture of the St. Lawrence and Ottawa Rivers). They obtained an additional decrease of TOC and $UV_{254}$ absorbance by 7.7 and 6.7%, respectively, when compared to the only coagulation process. Initial ozonation of the water reduced $UV_{254}$ absorbance more than TOC content, which was consistent with the results reported by Chiang et al. [9]. Supporting coagulation by ozonation and adsorption processes without and with UV irradiation allowed reducing $UV_{254}$ and $UV_{272}$ parameters by an additional 20%. A similar efficiency of $UV_{254}$ reduction was obtained by Tubić et al. [29] conducting initial ozonation before groundwater coagulation. Ozone reactions with humic compounds (fulvic acids predominate in surface water and are more reactive than humic acids) under neutral pH lead to a slight decrease in TOC content but a significant decrease in the value of UV absorbance. Such changes in the tested parameters indicated a decrease in the number of organic fractions with large molecular weights and an increase in the number of organic substances with lower molecular weights [1,30,31].

Wang et al. [5] obtained a reduction in DOC and UV254 absorbance by 45 and 51%, respectively, and in coagulation and adsorption applied later, by 76 and 81%, respectively, with both results obtained in the course of conducting research into the process of coagulation with polyaluminum chloride. In contrast, in coagulation preceded by ozone oxidation and finalized with adsorption, the reduction of DOC and $UV_{254}$ absorbance was 85 and 91%, respectively. These results were significantly better than those presented in this study (especially with regard to adsorption, however, carbon was not introduced during coagulation but after it, as a separate step of the process of water treatment).

The metal for which the highest reduction of its concentration in water was recorded as a result of the combined purification processes was lead (Table 7 and Figure 4). Coagulation with the use of PAX18 resulted in 50% removal of metal ions. Combining coagulation and adsorption with CWZ-22 powdered activated carbon resulted in an increase in the amount of metal removed by up to 63%. In the case of lead, additional ozonation of the sample had little impact on the efficiency of the process (53%), and using UV irradiation did not cause noticeable changes in the concentration of metal in the purified water. The usefulness of the sorption process in removing heavy metals from the water was often demonstrated. The adsorption process is usually characterized by high efficiency in terms of removing metals due to the fact that the mechanism combines the features of physical sorption, ion exchange, complexation, precipitation, as well as redox processes [22]. The authors provide the values of the efficiency of adsorption processes using various materials in a very wide range from a dozen to almost 100% [32]. The results obtained with the use of PAC fit this range.

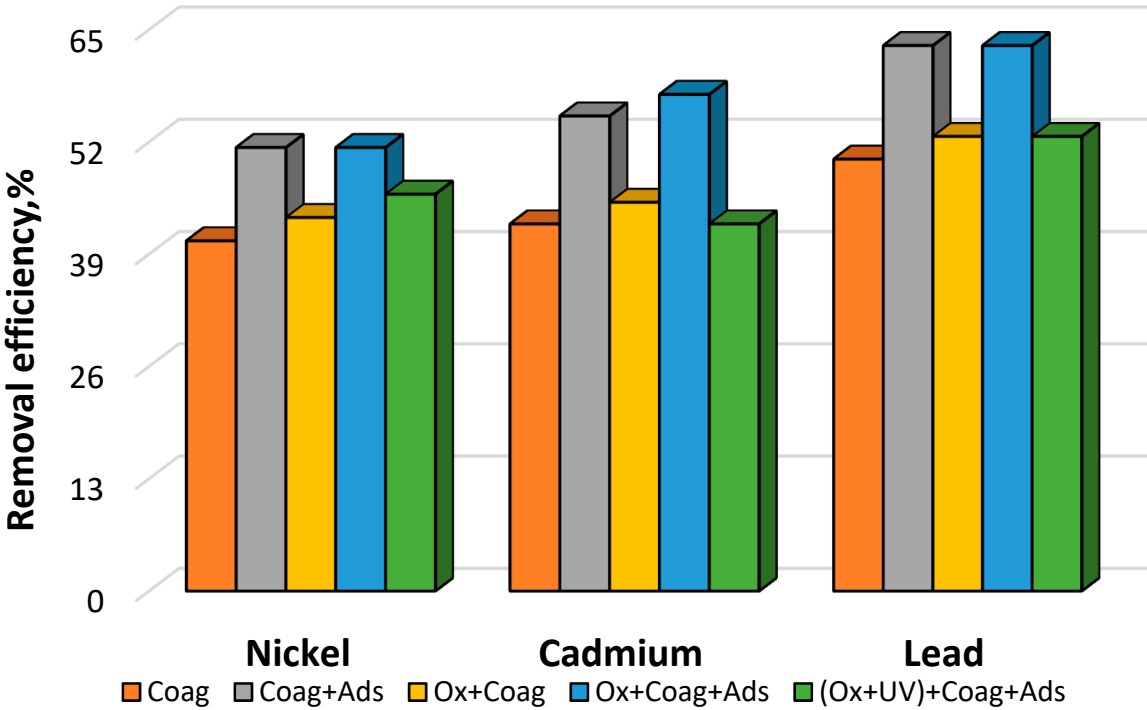

**Figure 4.** Heavy metals removal efficiency during coagulation (Coag), coagulation and adsorption (Coag + Ads), ozone oxidation and coagulation (Ox + Coag), ozone oxidation and coagulation and adsorption (Ox + Coag + Ads), ozone oxidation + UV irradiation and coagulation + adsorption ((Ox + UV) + Coag + Ads).

## 4. Summary

The presented work demonstrated the possibility of using pre-hydrolyzed polyaluminum chlorides for removing color, organic matter, as well as nickel, cadmium, and lead ions from surface water. Good water purification efficiency was achieved with both a low and high alkaline coagulant. It has been shown that supporting the coagulation process by adsorption with PAC, as well as additional ozonation, can improve the treatment processes of surface water for the purpose of supplying the population with water for consumption. Using processes associated with coagulation, to the greatest extent, increased the efficiency of removing color by 4–16% and reduced the value of $UV_{254}$ and $UV_{272}$ absorbance by 10–20% with regard to the use of coagulation as a unit process. An additional UV irradiation process was inefficient for the improvement of surface water treatment. If in the treatment of the tested, modified surface water, only one process supporting coagulation should be indicated, the best results were obtained with the involvement of additional adsorption.

Heavy metal ions such as nickel, cadmium, and lead are mostly removed by coagulation and adsorption processes, as well as combinations of the two. Additionally including ozone oxidation resulted in an insignificant decrease in the concentration of metal ions in the purified water.

During the treatment of the tested, modified surface water, the best efficiency was observed after the use of coagulation and adsorption processes enhanced with ozone oxidation. The additional involvement of UV irradiation did not have a significant effect on the removal of the analyzed pollutants.

**Author Contributions:** Conceptualization B.K. and E.S.; methodology, B.K. and E.S.; investigation, B.K. and E.S.; writing—original draft preparation, B.K. and E.S.; writing—review and editing, B.K. and E.S.; visualization, B.K. and E.S.; supervision, B.K.; funding acquisition, B.K. and E.S. All authors have read and agreed to the published version of the manuscript.

**Funding:** The scientific research was funded by the statute subvention of Czestochowa University of Technology, Faculty of Infrastructure and Environment, out within the research project No. BS/PB-400/301/2022.

**Data Availability Statement:** Not applicable.

**Acknowledgments:** Authors gratefully thank Lidia Dąbrowska at Czestochowa University of Technology for substantive discussions and her contribution to this work.

**Conflicts of Interest:** The authors declare no conflict of interest.

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
