# Peer review of "Organic Matter and Heavy Metal Ions Removal from Surface Water in Processes of Oxidation with Ozone, UV Irradiation, Coagulation and Adsorption"

_water, doi:10.3390/w14223763_

Round 1

Reviewer 1 Report

General Comments:

1.      The manuscript contains grammatical errors that should be corrected.

2.      The conclusive remarks are not supported by the experimental results.

 Specific Comments:

1.      Figure 1 shows that activated carbon was added to beakers no. 2, 4 and 5, whereas the text (line 158) states that activated carbon was added to the water in the beaker no. 2 and 4.   Please correct.

2.      Lines 170-179:  Please provide more details on the analytical methods. In particular, the operating conditions of instruments should be mentioned.

3.      Lines 215- 217:  Please provide experimental support for the statement on these lines. What types of aromatic compounds were present in the water? Did the authors analyze for the aromatic compounds before and after UV irradiation? Supporting results should be presented.

4.      The results presented in Table 6 are not consistent with the description in the text. Lines 295-297 mention reduction in the content of organic substances designated as OXI and TOC with the application of powdered activated carbon. However, Table 6 shows increases in the values of OXI, TOC and DOC of treated water compared to the values of modified surface water when using the adsorbents. This needs clarification.  The increase of organic content should be mentioned and the reasons behind this observation should be discussed. 

5.      Table 7:  In the Treated Water section of this table, there are two columns with the title of “Coag.”.  Please correct.

6.      The results presented in Table 7 and the text do not match. For example, Table 7 shows the value of 5 for color upon application of coagulation and the value of 12 for color after coagulation and adsorption.  This is not a reduction.  Similarly, Table 7 shows increase of color after coagulation, adsorption and oxidation. Please examine the reported values in table 7 and correct the text accordingly.

7.      The same comment as above applies to other parameters reported in Table 7. Coagulation alone (last column) shows the best result for Absorbance, NTU, OXI, DOC and TOC (except for (Ox+UV)+ Coag+ Ads).  This is confusing and unexpected.  The authors should more carefully examine the reported values.

8.      The reported values in Table 7 do not support the results presented in Figure 3.

9.      Lines 349-352:  The remarks on these lines are not supported by experimental results.  The authors should provide sufficient experimental support for the comments made.

Reviewer 2 Report

The manuscript “Organic matter and heavy metal ions removal from water in processes of oxidation with ozone, UV irradiation, coagulation and adsorption” showed good work. Some problems below should be addressed.

-Q1: The water temperature record during experiments should be added to the method. It controls the experimental results, such as the process that ozone involves.

-Q2: UV272 is mainly used to characterize the concentration of chlorine-containing DBPs. However, ozone especially causes bromate. UV272 linking to DBPs caused by ozone? The authors should add explanations to the manuscript.

-Q3: The detailed method or citations of measuring oxidisability should be added.

-Q4: Lines 184-185. Please check this sentence.

-Q5: The authors should add error bars in Figures and tables.

-Q6: Table7. The heavy metals have been removed, but the residual concentration still seems very dangerous. Please compare the final residual concentration of heavy metals with well-known water quality standards and discuss the comparison results.

Reviewer 3 Report

This work investigates the possibility of different combinations of processes: (a) coagulation and adsorption; (b) ozone oxidation and coagulation; (c) ozone oxidation, coagulation and adsorption; (d) ozone oxidation, UV irradiation, coagulation and adsorption for water treatment. And the performance of the related combined process was measured by changes in pH, turbidity, color, organic content, and heavy metal ions. The results of the study suggest that coagulation processes supported by adsorption with additional ozone and UV irradiation are beneficial to the water treatment process. The study has implications for the treatment of compound pollution in water and the development of new materials/new processes. However, the language and experimental design of the article can be further improved and perfected.

Some detailed comments are stated as follows:

1. The abstract is written in a way that lacks wholeness and does not provide a good summary of the whole paper; it also lacks science and does not have a clear scientific question to pose. The abstract should include the significance of the purpose of the study, the means and methods of the study, and the results and conclusions of the study to enhance logic. It is recommended that it be re-corrected.

2. The selection of keywords is too cumbersome, and the general number is recommended to be 4-5. The selection also needs to be able to reflect the core idea of this paper. It is recommended that it be re-corrected.

3. The specification of writing needs to be checked and improved. For example:

(1) Abbreviation formatting issues, e.g. "Ni2+, Cd2+ and Pb2+", "PAC". It is recommended that the full text be reviewed and amended.

(2) A space is needed between the number and the unit, e.g. "mgAl" in line 120 should be written as "mg Al". It is recommended that the full text be reviewed and amended.

(3) The format of the symbols needs to be standardized, e.g. whether spaces are needed on both sides of the "+" in the figure note "(Ox+UV)+Coag+Ads - ozone oxidation + UV irradiation and coagulation and adsorption" in Figure 1.

(4) Whether the "The UV254 and UV254 absorbance was equal to 0.298-365 and 0.245-296 1/cm, respectively." is written and expressed correctly.

(5) Valid values are recommended to be unified, e.g. "0.070 and 0.001 mg/L" and "0.38 to 0.40 mg/L".

(6) The inconsistent spaces at the beginning of paragraphs, e.g. the title of 3.2 is not the same as the other formats.

(7) Improper use of punctuation, e.g., line 253.

(8) The graphical data lacks error bars, e.g., Figure 2-4.

(9) The writing format of the unit needs to be checked.

(10) The whole manuscript should be polished.

4. The paragraphs of the introduction are too cumbersome, and it is recommended to merge some relevant paragraphs. The introduction of adsorption, coagulation and oxidation lacks strong arguments with practical examples. The introduction of the last paragraph is a bit inexplicable and does not suggest why this research work was done. It is suggested that the introduction be reconstructed.

5. In the "2.1 Materials" is recommended to add basic information about water quality. The water samples were taken for the time period December 2021 to February 2022, are they representative.

6. In "2.2 Course of research and analysis methodology", how to determine the experimental conditions such as experimental time, agent dosage, etc. What is the basis for the combination of processes in the experimental steps? Does the sequence of coagulation, adsorption and oxidation have an effect on the removal of pollutants? Please provide additional explanation.

7. In "3. Results and Discussion" with "3.1. Water ozonation" between the description of water quality, proposed in the second part. In "3.1. Water ozonation", for ozone + UV irradiation why 10 min was chosen as the reaction time, and whether the extended time can improve the removal rate of pollutants. Also, there is no specific explanation why increasing UV irradiation can improve the removal of organic matter but has no effect on turbidity and color?

8. The "3. Results and Discussion" gives the reader the impression of how effective the wastewater treatment is under what process conditions, remaining only with the description of the experimental phenomena and giving no detailed explanation. At the same time, there is no unique insight into the experimental results, no basis for the combined process is given, and the full text does not clearly reflect where the advantages of the combined process lie compared to the individual processes.

9. What are the advantages of this study compared with other methods, what are the engineering implications, and what are the problems that hinder its engineering application? The crude experimental investigation experiments in this paper cannot provide a good theoretical reference for its practical application. Please provide additional explanation.

10. And the concluding prospective section is a bit thin, and it can provide its own views on the issues raised in the paper and add specific suggestions for improvement instead of just repeating the experimental results in the paper again. The presentation of research findings needs to be more academic, percentages do not represent rigorous technical progress, and quantitative efficiency analysis is important.

Reviewer 4 Report

Title: Please redo the title: It should clearly mention type of water for readers to instantly pick the reading interest for this paper as author used “Surface Water”.

Abstract: Needs improvement especially the conclusion part is not stated properly or missing.

Introduction:

Line 48: Use NOM onwards where necessary 

Line 49: Trihalomethanes (THMs) and haloacetic acids (HAAs) consistent with given terms.

Line 59: Write complete term before abbreviation for PAC.

Line 59-61: Reference is missing.

Line 62-71: Not clear seems repetition also authors are encouraging to add citation (s) to support the literature.

Line 85: Citation is missing.

Line 94: pH range is missing for the highest removal of mentioned metals? Authors are instructed to keep visibility and consistency in write-up.

Line 101: Report? Clear the term.

General comments:

The reason for choosing specific heavy metals Ni, Cd and Pb is not clearly stated.

Background on Activated Carbon is missing authors can add some potentials of using Activated Carbon for present study.

Research Materials, Course and Methodology

Line 131: Keep clarity as Working volume of 1.7 L.

Line 145 + 161: Sedimented change it to settling time.

Line 150: A filter paper specification? Like filter paper brand and number.

Line 154: UV irradiation with 254nm and 272nm is missing? Not clearly stated please verify this stage. Also, in figure one 272nm is not mentioned. However, it was written for both.

Line 160: Replace fast mixing with Rapid Mixing.

General Comments:

Authors can split Figure 1 in each particular stage respectively, followed to perform the experimental work in order to see the visibility at each stage of process performed.

Please mention if any standards like ASTM/EPA/ or any other relevant which are used to perform the experimental work.

Specification of Al used in this manuscript can be also added.

Results and Discussion

Line 181-182: Rephrase action required.

Line 186-187: Please state the standards followed to justify the range of acquired results.

Line 194-197: Please rephrase and state with clarity.

General comments:

Results need more justification. Lack of references can be seen; authors are encouraged to support results with more related work to compare the 4 stages outcome as best removal in respective sections.

Summary

General comments: Summary needs more elaboration on potential use of AL and PAC for the successful removal pollutants. 

Round 2

Reviewer 1 Report

The quality of manuscript has improved after the revision and it can be published after minor revisions, as indicated below:

-          Please comment on the results in Table 7 that show a higher efficiency of turbidity (NTU) removal (lower NTU value) with coagulation only compared to coagulation+adsorption and coagulation+adsorption+oxidation.

Author Response

Reviewer 1 comments

The quality of manuscript has improved after the revision and it can be published after minor revisions, as indicated below:

- Please comment on the results in Table 7 that show a higher efficiency of turbidity (NTU) removal (lower NTU value) with coagulation only compared to coagulation+adsorption and coagulation+adsorption+oxidation.

Thank you very much for the recommendation. We would like to explain that a slight decrease of turbidity removal compared to coagulation alone was a result of the penetration of fine PAC particles into the treated water. Samples were decanted, not passed through the filter paper. However, it should be noted that in practice, under technical conditions, the coagulation process is always followed by a filtration process in which PAC particles would be separated from the purified water.

The additional information was introduced to the manuscript. We would like to thank the Reviewer for the work put into the review of our report. All comments, suggestions and recommendations allowed us to look at our manuscript objectively and enriched its final version.

Reviewer 2 Report

Comments were well responded, and this manuscript can be accepted.

Author Response

We would like to thank the Reviewer for the work put into the review of our report. All comments, suggestions and recommendations allowed us to look at our manuscript objectively and enriched its final version. Thank you very much again.

Reviewer 3 Report

None

Author Response

(The authors gave the same response as above.)
